# Green Transformation of Anti-Epidemic Supplies in the Post-Pandemic Era: An Evolutionary Approach

**DOI:** 10.3390/ijerph19106011

**Published:** 2022-05-15

**Authors:** Han Xiao, Cheng Ma, Hongwei Gao, Ye Gao, Yang Xue

**Affiliations:** 1School of Business, Qingdao University, Qingdao 266071, China; xao1299@163.com (H.X.); mc_0812@163.com (C.M.); gaoye1988@126.com (Y.G.); 2School of Mathematics and Statistics, Qingdao University, Qingdao 266071, China; 3The Center for Data Science in Health and Medicine, Qingdao University, Qingdao 266071, China

**Keywords:** green transformation, evolutionary game, COVID-19, government regulation, environmentally friendly masks

## Abstract

Post-pandemic, the use of medical supplies, such as masks, for epidemic prevention remains high. The explosive growth of medical waste during the COVID-19 pandemic has caused significant environmental problems. To alleviate this, environment-friendly epidemic prevention measures should be developed, used, and promoted. However, contradictions exist between governments, production enterprises, and medical institutions regarding the green transformation of anti-epidemic supplies. Consequently, this study aimed to investigate how to effectively guide the green transformation. Concerning masks, a tripartite evolutionary game model, consisting of governments, mask enterprises, and medical institutions, was established for the supervision of mask production and use, boundary conditions of evolutionary stabilization strategies and government regulations were analyzed, and a dynamic system model was used for the simulation analysis. This analysis revealed that the only tripartite evolutionary stability strategy is for governments to deregulate mask production, enterprises to increase eco-friendly mask production, and medical institutions to use these masks. From the comprehensive analysis, a few important findings are obtained. First, government regulation can promote the green transformation process of anti-epidemic supplies. Government should realize the green transformation of anti-epidemic supplies immediately in order to avoid severe reputation damage. Second, external parameter changes can significantly impact the strategy selection process of all players. Interestingly, it is further found that the cost benefit for using environmentally friendly masks has a great influence on whether green transformation can be achieved. Consequently, the government should establish a favorable marketplace for, and promote the development of, inexpensive, high-quality, and effective environmentally friendly masks in order to achieve the ultimate goal of green transformation of anti-epidemic supplies in the post-pandemic era.

## 1. Introduction

Since the outbreak of the COVID-19 pandemic in 2020, with several iterations of the epidemic, non-pharmaceutical interventions, such as indoor facemasks, lateral-flow testing, etc., have been studied and recognized by researchers and used by governments [1]. Therefore, anti-epidemic supplies, particularly masks, have been in high demand [2], and wearing masks has become the norm [3]. Polypropylene and high-density polyethylene are the primary raw materials used for most masks on the market [4]. Regardless of the treatment method adopted, such materials pollute the environment, causing soil, water, and microplastic pollution [5,6]. Discarded masks are classified as medical waste, and their treatment method is distinct from that of other pollutants. Most countries consider safety as the primary criterion for the treatment of medical waste. The United States, Europe, and other countries have focused on medical waste and unified its treatment. In 2003, owing to the outbreak of severe acute respiratory syndrome (SARS), China issued the “Regulations on the Administration of Medical Wastes” that stipulate that “medical waste must be transported separately (no railways or air transportation is allowed), and the temporary storage time for medical waste must not exceed two days” [7]. Following the outbreak of COVID-19 in 2020, medical waste treatment units in seriously affected areas were unable to operate normally. In 2020, the Ministry of Ecology and Environment of the People’s Republic of China issued the “Management and Technical Guidelines for Emergency Disposal of Medical Waste in COVID-19” (Trial Implementation). The guidelines allow processing departments to perform emergency treatment during the epidemic period, and they are exempt from environmental impact assessment, medical waste business licenses, and other procedures [8]. This measure allowed more medical waste to be disposed of on time, easing the pressure on medical institutions and medical waste treatment enterprises. However, the ensuing question is whether medical waste that has not been subjected to an environmental assessment is causing long-term and serious damage to the environment. The pandemic’s environmental problems affect not only China, but the entire world. In 2020, Oceans Asia, a marine protection organization, released a report stating that about 52 billion masks were manufactured globally in 2020, and approximately 1.56 billion masks were discarded in the sea, resulting in an additional 4680 to 6240 cubic metric tons of marine plastic pollution. This polluting waste is estimated to require 450 years for decomposition. Most of the remaining 50.5 billion masks were burned as medical waste, and some were buried, discarded, or otherwise disposed of [9].

To alleviate the environmental deterioration caused by the sharp increase in medical waste, researchers have explored new ways to effectively reduce the pollution from medical waste, and it has been shown that reusable masks have a positive effect on reducing the environmental burden. If properly disinfected, reusable masks can maintain the same filtering effect as disposable masks [10,11]. Moreover, the price of reusable masks is lower than that of ordinary disposable masks, and they have less impact on climate change [12]. A single reusable mask can be effectively used 183 times, and doing so would reduce the CO_2_ emission from 0.06 kg/piece to 0.036 kg/piece, along with other pollutants [13]. Lee et al. [14] discovered that using an embedded filtration layer (EFL) reduces emissions to the atmosphere by 0.25 kg/piece, while the waste generated is only 0.0004 kg/piece, or one-tenth of that of disposable masks. Imbrie-Moore et al. [15], Swennen et al. [16], and Thomas et al. [17] studied the production of reusable masks or filter elements using 3D-printing technology, and showed that when the filter element is sterilized or replaced correctly, the reusable mask not only has the same performance as a disposable mask, but can also be produced faster; this can effectively improve the shortage of protective materials. Furthermore, reusable masks combined with cardiopulmonary resuscitation masks [18] and the new generation of self-sterilizer masks developed using intelligent nanomaterials [19], collectively referred to as environmentally friendly masks (EFMs) hereafter, are new ideas for the development of reusable mask technology that have been confirmed through experimental verification.

With the continuous development of EFMs, some countries have recognized their economic and environmental advantages and have begun to produce and promote them. The EFL reusable face mask developed by Singapore Forever Family Co., Ltd. (Singapore) can be reused 30 times and is certified by EN14683 and ASTMF2101, meaning that this type of reusable face mask has the same performance characteristics as disposable surgical masks [14,20]. The Hong Kong Research Institute of Textile and Apparel developed a copper core anti-epidemic mask called CuMask+, that can be reused for 60 days, and 30 million masks were distributed to the public by late June 2020 [21].

However, the promotion process of EFMs is not smooth. Scholars have found that, owing to the influence of cleaning, price, trust, and other factors, consumers are prone to form negative attitudes toward EFMs. Moreover, owing to the relatively complex washing procedure, not all consumers are willing to use EFMs. Incorrect cleaning methods can destroy EFMs’ protective performance and increase the risk of viral infection [22]. Furthermore, it is difficult for medical product recycling to overcome cost barriers. Given that the concern for safety drives various decisions of medical institutions, some recyclable products are discarded in advance, and the value of recycling is much lower than expected [23]. Lack of trust in consumer groups is also a problem faced in EFM promotion. The government’s decision to make the CuMask+ available to the public for free has sparked widespread criticism from the public and mainstream media. When there is no immediate need for masks, people believe that a reusable mask with a higher government procurement price is questionable [21].

In response to the difficulties encountered in the promotion process, scholars have suggested that government regulation is an important solution to consumers’ possible negative attitudes toward EFMs. Makki et al. [24] believe that under appropriate reward and punishment measures, there will be alternative and sustainable consumer behavior; that is, consumers will buy masks made of eco-friendly raw materials. Kim et al. [25] pointed out that, although economic benefits or incentives can change acceptance, non-monetary rewards have potential advantages over monetary compensation, and the government or organizations should adopt appropriate incentives depending on the circumstances. Appropriate education, supervision, motivation, and punishment can alleviate these negative attitudes. Opole General Hospital in Poland enables staff to classify infectious and non-infectious medical waste more accurately through training and encouragement, which can reduce waste by half, and waste disposal costs by 79% [23].

Government regulations not only directly affect consumer behavior, but also affect enterprise production behavior, indirectly affecting consumer behavior and reducing environmental pollution. Environmental regulation is beneficial to improve the green technology innovation level and to promote and upgrade green transformation [26,27,28,29]. Frewer et al. [30] demonstrated that public acceptance of dangerous goods depends on self-perceived benefits. When the public believes that the benefits of dangerous goods are sufficiently large, negative attitudes are reduced accordingly. Governments can promote business innovation by subsidizing the research and development of innovative products, reducing costs, and improving product quality [31]. This is beneficial in terms of lowering the selling price and reducing consumer dissatisfaction due to high prices. Bansal and Gangopadhyay [32] studied the impact of unified or discriminatory state policies to reduce pollution and found that, while subsidies can improve environmental quality, tax policies can worsen it. Hafezalkotob [33] considered the competition between green and conventional supply chains under six scenarios and found that the government has a significant impact on the interests of the supply chain and the environment, and should formulate reasonable reward and punishment regulations to ensure people’s health. Safarzadeh and Rasti-Barzoki [34] established a multi-stage game model under the two scenarios of government tax cuts and subsidies and demonstrated that tax cuts are more effective in achieving sustainable transformation.

After reviewing the above literature, this paper found that in reality, there are contradictions among various stakeholders in the use, production, and promotion of EFMs. These contradictions cause stakeholders to interact with each other in the selection process, and it is necessary to continuously adjust and improve their strategic choices to gradually reach a balanced state. This paper attempts to address the following three questions:What is the impact of government regulations on green transformation?How do enterprises and medical institutions make decisions in the presence of government regulations?What are the factors influencing the green transformation?

To address these questions, this paper proposed an evolutionary game theory based on bounded rationality. Evolutionary game groups gradually achieve stable strategy equilibrium through processes such as selection, imitation, or mutation that can describe the changing trend of group behavior, predict individual strategy choices and final states, and study and discuss the behavior of interest groups [35,36]. Evolutionary game methods are widely used in medical and green-transformation research. Tooby et al. [37] and Yong and Choy [38] used evolutionary game research to discuss the behavior of people who were ignoring safety measures and enjoying the convenience brought by others’ compliance with the system during the epidemic, proposing the formulation of effective reward and punishment measures to encourage compliance with laws and regulations. Kabir et al. [39] analyzed mask-wearing behavior and quantified the use of masks and other protective behaviors. Xu et al. [40] used a tripartite evolutionary model to analyze the impact of different factors on the decision-making of participants in public health emergencies. Fu et al. [41] used evolutionary games to simulate vaccinations. Iwamura et al. [42] described the decision-making process for vaccination by introducing self-protection as a new strategy.

In the study of green transformation, Sun et al. [43] used an evolutionary game model to predict green investment strategies under three government subsidies and found that these subsidies can reduce free-riding behavior. By establishing an evolutionary game model of the Indian government and textile enterprises in three different scenarios, Mahmoudi and Rasti-Barzoki [44] proposed that the imposition of tariffs could effectively reduce the impact of enterprises on the environment. Xu et al. [45] and Zhu and Dou [46] used an evolutionary game model to describe the interactions among suppliers, manufacturers, and governments for green supply chains and government regulations in which the government environmental policy is optimized to encourage green activities. Li and Zhou [47] proposed that government regulation can promote innovation in green technologies, and consolidate the level of cooperation between universities, enterprises, and governments. Cui et al. [48] established an evolutionary game model in the context of green financial policies. The study indicated that the government will encourage financial institutions to choose green financial services and that consumers’ purchase intentions depend on the consumer product benefits. Huang et al. [49] proposed strategies to promote the development of shared green industries by comparing the processes of green industry investment and resource sharing. Sun et al. [50] built an evolutionary game model for ecotourism development, and punishment had a greater impact on players than incentives, with cost being the key factor affecting strategy.

The COVID-19 pandemic will continue, even if most people are vaccinated [51]. The conflicts between medical and environmental protection need to be solved. Considering the environmental problems caused by epidemic prevention supplies, this study used a more realistic method to analyze and establish an evolutionary game model for governments, enterprises, and medical institutions in various places. Medical institutions were selected as the EFMs consumer group because they use a large number of masks and are easier to manage than individuals. The influence of the three players was studied, a replica dynamic system of an asymmetric evolutionary game was constructed, and a dynamic simulation analysis was conducted. Research has shown that the value of external variables directly affects the strategic choices of all players, and the only tripartite evolutionary stability strategy is for governments to deregulate mask production, enterprises to increase eco-friendly mask production, and medical institutions to use these masks. Furthermore, there were two interesting findings during the simulation process. On the one hand, liquidated damages imposed by enterprises on medical institutions for terminating the purchase of ordinary masks will have a negative impact on green transformation. Medical institutions, on the other hand, perceive green transformation negatively when the cost of using EFMs is the same as, or even higher than, that of disposable masks. No matter how much the government subsidizes it, green transformation will be hard to complete. The model established in this study and the main conclusions obtained can provide a basis for government regulations and a reference for relevant studies.

## 2. Evolutionary Game Model of the Green Transformation of Anti-Epidemic Supplies 

Promoting green transformation requires joint efforts by governments, enterprises, medical institutions, and market consumers. However, the process of green transformation is easily affected by factors such as irrationality and information asymmetry among groups or individuals. Simultaneously, there are significant differences in interest appeals between different stakeholders. To predict equilibrium states in unstable environments, Smith and Price [52] proposed the evolutionary game theory and the evolutionary stability strategy (ESS). The percentage of individuals who choose different pure strategies in the population is used to replace mixed strategies in the evolutionary game theory, which is based on the biological evolution theory.

This study adopted an evolutionary game approach, considered the repeated interactions of multiple enterprises, medical institutions, and governments, and predicted tripartite evolutionary stability strategies. Government strategies include regulating and deregulating the use and production of EFMs, abbreviated as “regulate” and “deregulate,” respectively. The strategies of enterprises include increasing and not increasing the production of EFMs, abbreviated as “increase production” and “no production increase,“ respectively. The strategies of medical institutions include using EFMs, and using disposable medical masks or disposable regular masks, abbreviated as “use EFMs” and “use regular mask,” respectively.

The study made the following assumptions:Compared with regular masks, EFMs have the same indicators, that is, the protection ability and comfort are the same, except for the difference in reusability, cleaning, and replacement.The number of masks used daily by medical institutions is fixed. Therefore, the increase in the use of EFMs is equal to the reduction in the use of regular masks.The market price of any type of mask does not change with the increase or decrease in supply or demand.The difference between the cost of detergent, electricity, and labor, and the advantage of reusable EFMs, is the cost benefit of these masks. The cost benefit is positive if the savings from using EFMs exceed the cost of reuse. Otherwise, the cost benefit is considered negative [53,54].Liquidated damages for the termination of medical institutions’ purchase of ordinary masks imply that medical institutions have long-term purchase orders for regular masks. If medical institutions reduce orders for regular masks, they must pay liquidated damages [55].

The specific model parameters and the evolutionary game payoff matrix are shown in Table 1 and Table 2, respectively.

## 3. Stability Strategy Analysis of the Evolutionary Process

### 3.1. The Replicator Dynamics Equation of Governments, Enterprises, and Medical Institutions

#### 3.1.1. The Replicator Dynamics Equation of Governments

Consider that the proportion of government regulations is x. Then, the proportion of deregulation is 1−x. The expected benefits of the two government initiatives are $U_{1x}$ and
$U_{2x}$. The total benefit to the government group is ${\bar{U}}_{x}$, then,
(1)$$\begin{array}{cc} U_{1x} & =yz\left(R-G_{c}-G_{H}-C_{G}\right)q+y\left(1-z\right)\left(f_{1}-G_{c}-C_{G}\right)q+\left(1-y\right)z\left(R-G_{H}-C_{G}\right)q\\ & +\left(1-y\right)\left(1-z\right)\left(f_{1}-C_{G}\right)q \end{array}$$
(2)$$U_{2x}=yz\left(R-H\right)q-y\left(1-z\right)\left(Hq\right)+\left(1-y\right)z\left(R-H\right)q-\left(1-y\right)\left(1-z\right)Hq$$
(3)$${\bar{U}}_{x}=xU_{1x}+\left(1-x\right)U_{2x}$$

According to the expected payoff expression of governments, the replicator dynamic equation of governments is:(4)$$F\left(x\right)=\frac{dx}{dt}=x\left(1-x\right)\left(U_{1x}-U_{2x}\right)=x\left(1-x\right)\left(H+f_{1}-C_{G}-zG_{H}-zf_{1}-yG_{C}\right)q$$

The influence of *x* on the evolutionary stable equilibrium strategy of all governments is calculated as:(5)$$\frac{dF\left(x\right)}{dx}=\left(1-2x\right)\left(H+f_{1}-C_{G}-zG_{H}-zf_{1}-yG_{C}\right)q$$

#### 3.1.2. The Replicator Dynamics Equation of Enterprises

Consider that the proportion of enterprises increasing production is y. Then, the proportion of enterprises not increasing production is 1−y. The expected benefits of the two initiatives of enterprises are $U_{1y}$ and $U_{2y}$. The total benefit to the enterprises group is ${\bar{U}}_{y}$, then,
(6)$$\begin{array}{cc} U_{1y} & =xz\left(\alpha P+G_{c}-\beta C-C_{S}\right)q+x\left(1-z\right)\left(G_{c}-\beta C-C_{S}\right)q\\ & +\left(1-x\right)z\left(\alpha P-\beta C-C_{S}\right)q+\left(1-x\right)\left(1-z\right)\left(-\beta C-C_{S}\right)q \end{array}$$
(7)$$U_{2y}=xz\left(-P+f_{2}\right)q+\left(1-x\right)z\left(-P+f_{2}\right)q$$
(8)$${\bar{U}}_{y}=yU_{1y}+\left(1-y\right)U_{2y}$$

According to the expected payoff expression of enterprises, the replicator dynamic equation of enterprises is: (9)$$F(y)=\frac{dy}{dt}=y\left(1-y\right)\left(U_{1y}-U_{2y}\right)=y\left(1-y\right)(xG_{C}+z\alpha P-zf_{2}+P-\beta C-C_{S})q$$

The influence of y on the evolutionary stable equilibrium strategy of all enterprises is calculated as: (10)$$\frac{dF\left(y\right)}{dy}=\left(1-2y\right)\left(xG_{C}+z\alpha P-zf_{2}+P-\beta C-C_{S}\right)q$$

#### 3.1.3. The Replicator Dynamics Equation of Medical Institutions

Consider that the proportion of medical institutions that use EFMs is z. Then, the proportion of institutions using regular masks is 1−z. The expected benefits of the two initiatives of medical institutions are $U_{1z}$ and $U_{2z}$. The total benefit to medical institutions group is ${\bar{U}}_{z}$, then,
(11)$$\begin{array}{cc} U_{1z} & =xy\left(G_{H}-\alpha P-d+r_{c}\right)q+x\left(1-y\right)\left(G_{H}-\alpha P-d+r_{c}-f_{2}\right)q\\ & +\left(1-x\right)y\left(-\alpha P-d+r_{c}\right)q+\left(1-x\right)\left(1-y\right)\left(-\alpha P-d+r_{c}-f_{2}\right)q \end{array}$$
(12)$$U_{2z}=xy\left(-f_{1}\right)q+x\left(1-y\right)\left(-f_{1}\right)q$$
(13)$${\bar{U}}_{z}=zU_{1z}+\left(1-z\right)U_{2z}$$

According to the expected payoff expression of medical institutions, the replicator dynamic equation of medical institutions is
(14)$$F\left(z\right)=\frac{dz}{dt}=z\left(1-z\right)\left(U_{1z}-U_{2z}\right)=z\left(1-z\right)\left(xG_{H}+xf_{1}+yf_{2}-\alpha P-d+r_{c}-f_{2}\right)q$$

The influence of *z* on the evolutionary stable equilibrium strategy of all medical institutions is calculated as
(15)$$\frac{dF\left(z\right)}{dz}=\left(1-2z\right)\left(xG_{H}+xf_{1}+yf_{2}-\alpha P-d+r_{c}-f_{2}\right)q$$

### 3.2. Evolutionary Stability Strategies

**Proposition** **1.***The proportion of government regulation decreases with an increase in the proportion of enterprises that have increased EFM production*.

**Proof.** Let $\lambda_{y}=\frac{H+f_{1}-C_{G}-zG_{H}-zf_{1}}{GC}$.When $y>\lambda_{y}$, $\frac{dF(x)}{dx}|x=0<0$, the evolutionary stabilization strategy of governments is $x^{*}=0$, that is, when the proportion of enterprises that have increased EFM production is relatively large, $y>\lambda_{y}$, governments prefer deregulation. When $y=\lambda_{y}$, any proportion of government regulation is an evolutionary stability strategy. When $y<\lambda_{y}$, $\frac{dF(x)}{dx}|x=1<0$, the evolutionary stabilization strategy of governments is $x^{*}=1$, that is, when the proportion of enterprises that have increased EFM production is relatively small, $y<\lambda_{y}$, governments prefer regulation. □

**Proposition** **2.***The proportion of government regulation decreases with an increase in the proportion of medical institutions using EFMs*.

**Proof.** Let $\lambda_{z}=\frac{H+f_{1}-C_{G}-yG_{C}}{f_{1}+G_{H}}$.When $z>\lambda_{z}$, $\frac{dF(x)}{dx}|x=0<0$, the evolutionary stabilization strategy of governments is $x^{*}=0$, that is, when the proportion of medical institutions using EFMs is relatively large, $z>\lambda_{z}$, governments prefer deregulation.When $z=\lambda_{z}$, any proportion of government regulation is an evolutionary stability strategy.When $z<\lambda_{z}$, $\frac{dF(x)}{dx}|x=1<0$, the evolutionary stabilization strategy of governments is $x^{*}=1$, that is, when the proportion of medical institutions using EFMs is relatively small, $z<\lambda_{z}$, governments prefer regulation. □ 

Similarly, obtain Proposition 3 and Proposition 4.

**Proposition** **3.***The proportion of enterprises that increase EFM production increases with an increase in the proportion of government regulations and the proportion of medical institutions using EFMs*.

**Proposition** **4.***The proportion of medical institutions using EFMs increases as the proportion of government regulations and the proportion of enterprises that produce EFMs increase*.

**Proposition** **5.***When reputation loss is greater than the critical reputation loss*$C_{G}+G_{C}+G_{H}-G_{C}(1-y)$*, the government group will choose to regulate, regardless of the decisions made by medical institutions. When reputation loss is greater than the critical reputation loss*$C_{G}+G_{C}+G_{H}-(G_{H}+f_{1})(1-z)$*, the government group will choose to regulate, regardless of the decisions made by enterprises*.

**Proof.** Because $\lambda_{z}=\frac{H+f_{1}-C_{G}-yG_{C}}{f_{1}+G_{H}}=1+\frac{H-C_{G}-G_{C}-G_{H}}{f_{1}+G_{H}}+\frac{G_{C}(1-y)}{f_{1}+G_{H}}$ and 0<y<1, so $\frac{G_{C}(1-y)}{f_{1}+G_{H}}>0$.When $H-C_{G}-G_{C}-G_{H}\geq -G_{C}(1-y)$, it can be concluded that $0<y<1<\lambda_{z}$. Therefore, when the difference between reputation damage from government deregulation and regulation cost is greater than $-G_{C}(1-y)$, the proportion of government regulation tends to 1, regardless of the decisions made by enterprises.Setting $C_{G}+G_{C}+G_{H}-G_{C}(1-y)$ as a critical reputation loss, when reputation loss is greater than critical reputation loss, the government group will choose to regulate, regardless of the decisions made by medical institutions.Similarly, get
$$\lambda_{y}=\frac{H+f_{1}-C_{G}-zG_{H}-zf_{1}}{G_{C}}=1+\frac{H-C_{G}-G_{H}-G_{C}}{G_{C}}+\frac{\left(G_{H}+f_{1})(1-z\right)}{G_{C}}$$Setting $C_{G}+G_{C}+G_{H}-(G_{H}+f_{1})(1-z)$ as critical reputation loss, when reputation loss is greater than the critical reputation loss, the government group will choose to regulate, regardless of the decisions made by enterprises. □

**Proposition** **6.***When reputation loss is less than the critical reputation loss*$C_{G}+G_{C}+G_{H}-G_{C}(1-y)$*, the proportion of government regulations increases as reputation damage increases. The proportion of government regulations decreases as government regulation costs and subsidies increase*.

When reputation loss is less than the critical reputation loss $C_{G}+G_{C}+G_{H}-(G_{H}+f_{1})(1-z)$, the proportion of government regulations increases as reputation damage increases. It also increases with an increase in the environmental penalty of medical institutions. The proportion of government regulations decreases with an increase in government regulation costs and subsidies for medical institutions.

The proportion of enterprises producing EFMs decreases with an increase in liquidated damages for the termination of medical institutions’ purchase of ordinary masks, EFM production machine transformation costs, and the average cost of EFMs. On the contrary, the proportion of enterprises producing EFMs increases with an increase in the average price of EFMs and subsidies for enterprises.

The proportion of medical institutions using EFMs decreases with an increase in the average price of EFMs, medical institution management cost, liquidated damages for the termination of medical institutions’ purchase of ordinary masks, and the environmental penalty of medical institutions. The proportion of medical institutions using EFMs and the cost benefit for using EFMs increases as subsidies increase.

**Proof.** When $H-C_{G}-G_{C}-G_{H}<-G_{C}(1-y)<0$, then $\lambda_{z}<1$, let y=1, have $\lambda_{z}=\frac{H+f_{1}-C_{G}-G_{C}}{f_{1}+G_{H}}$, let y=0, have $\lambda_{z}=\frac{H+f_{1}-C_{G}}{f_{1}+G_{H}}$. □

As shown in Figure 1, region 1 is not regulated and region 2 is regulated. Similarly, obtain Figure 2, Figure 3, Figure 4, Figure 5 and Figure 6, presented as follows:

In conclusion, The proportion of government regulation:(16)$$V_{11}=\frac{2H+2f_{1}-2C_{G}-G_{C}}{2f_{1}+2G_{H}}$$
(17)$$V_{12}=\frac{2H+f_{1}-C_{G}-G_{H}}{2G_{C}}$$

The proportion of enterprises increasing production,
(18)$$W_{21}=1-\frac{f_{2}+2\beta C+2C_{S}-2P-\alpha P}{2G_{C}}=\frac{2G_{C}-f_{2}-2\beta C-2C_{S}+2P+\alpha P}{2G_{C}}$$
(19)$$W_{22}=1-\frac{2\beta C+2C_{S}-G_{C}-2P}{2\alpha P-2f_{2}}=\frac{2\alpha P-2f_{2}-2\beta C-2C_{S}+G_{C}+2P}{2\alpha P-2f_{2}}$$

The proportion of enterprises increasing production,
(20)$$W_{31}=1-\frac{2\alpha P+2d+2r_{c}+f_{2}}{2G_{H}+2f_{1}}=\frac{2G_{H}+2f-2\alpha P-2d+2r_{c}-f_{2}}{2G_{H}+2f_{1}}$$
(21)$$W_{32}=1-\frac{2\alpha P+2d+2r_{c}+2f_{2}-G_{H}-f_{1}}{2f_{2}}=\frac{2f_{2}-2\alpha P-2d+2r_{c}-2f_{2}+G_{H}+f_{1}}{2f_{2}}$$

The influence of each parameter on the decision of each group is obtained using Equations (16)–(21), as shown in Table 3.

## 4. System Dynamics Simulation Analysis

System dynamics simulation is a quantitative feedback method for complex economic systems that uses computer simulation technology to test the accuracy and effectiveness of evolutionary game models through numerical simulations. In this model, this paper used system dynamics simulation analysis to study the impact of variables, such as cost and price, on evolutionary stability strategies to determine the impact of the strategic interaction among governments, enterprises, and medical institutions on the EFMs industry and to test the validity of the model.

### 4.1. System Dynamics Evolutionary Game Model

Government-related variables are: the environmental benefits of using EFMs R, reputation damage from government inaction H, subsidies for producing EFMs $G_{C}$, subsidies for using EFMs $G_{H}$, government regulation costs $C_{G}$, and environmental penalty for not using EFMs in medical institutions $f_{1}$.Enterprise-related variables are: the average price of a regular mask P, the average cost of a regular mask C, EFM production machine transformation cost $C_{S}$, price increase coefficient of EFMs α, and cost increase coefficient of EFMs β.The variables related to medical institutions are: medical institution management cost d, the cost benefit for using EFMs $r_{c}$, liquidated damages for the termination of medical institutions’ purchase of ordinary masks $f_{2}$, and the number of EFMs used q.

We drew a flow chart of the system to provide a quantitative analysis of the game players, as shown in Figure 7. It shows the relationship between variables.

### 4.2. Formatting of Mathematical Components

The dynamic equation of the system is established according to each variable. We determined the functional relationship between various variables in the system and analyzed the behavioral trends of each subject and the impact of policy changes on the system. This study used data as close to real life or actual data as possible to assign values to the exogenous variables in Table 1. Governments, enterprises, and medical institutions eventually reach an evolutionary stable strategy equilibrium through dynamic evolution.

Simulation analysis was performed using the Vensim system dynamics modeling software, setting parameters as: INITIAL TIME = 0, FINAL TIME = 15, and TIME STEP = 0.05. A simulation analysis of each influencing factor was conducted using the strategic ratios of governments, enterprises, and medical institutions as the main measurement indicators. From the above analysis of the dynamic equation of replication, eight pure strategy combinations can be observed: (0, 0, 0), (1, 0, 0), (0, 1, 0), (0, 0, 1), (1, 1, 0), (1, 0, 1), (0, 1, 1), and (1, 1, 1). A stable equilibrium was obtained using the Jacobian matrix.

The Jacobian matrix for government regulation of medical institutions using EFMs and enterprises to increase EFM production is as follows:$$J_{G}=\left(\begin{array}{ccc} \left(1-2x\right)\left(H+f_{1}-C_{G}-zG_{H}-zf_{1}-yG_{C}\right)q & -x\left(1-x\right)G_{C}q & -x\left(1-x\right)f_{1}q\\ y\left(1-y\right)G_{C}q & \left(1-2y\right)\left(xG_{C}+z\alpha P+P-zf_{2}-\beta C-C_{S}\right)q & 2y\left(1-y\right)\alpha P\\ z\left(1-z\right)\left(G_{H}+f_{1}\right)q & z\left(1-z\right)f_{2}q & \left(1-2z\right)\left(xG_{H}+xf_{1}+yf_{2}-\alpha P-d+r_{c}-f_{2}\right)q \end{array}\right)$$

The eigenvalues of each equilibrium point and attributes of the corresponding points are listed in Table 4.

The ESS point is $O_{7}$(0, 1, 1). The evolutionary game starts at (0.1, 0.1, 0.1) and eventually evolves to the equilibrium point $O_{7}$(0, 1, 1), as shown in Figure 8. (If the curve is closer to 1, it means that the probability of selecting regulation, increasing the production of EFMs, or using EFMs is higher, and vice versa. The following figure is similar.)

The government’s emphasis on regulation in the early stages of the green transformation of anti-epidemic supplies can promote cooperation among various groups. In the middle of the process, the proportion of regulations peaks and then decreases. Even if the government cancels relevant regulatory policies at a later stage (red dashed line in Figure 8), medical institutions and enterprises will evolve to use and produce EFMs (blue line and green dashed line in Figure 8). The main reason is that in the early days of the green transformation, medical institutions will rapidly increase the use of EFMs under the government’s regulations and subsidies; this will encourage enterprises to gradually increase their production of EFMs. Finally, the green transformation process will achieve a sustainable development state of epidemic prevention materials without government supervision.

Simultaneously, in the simulation process, it was found that the proportion of enterprises producing EFMs in the initial stage decreased slightly (blue line in Figure 8). This is because when medical institutions use fewer EFMs, production of more EFMs leads to greater losses, reducing the incentive to increase EFM production, therefore reducing the proportion of EFMs produced. This suggested that the formulation and implementation of regulations need to continue for a certain period before it can be judged whether they are effective. A government’s formulation of long-term regulatory policies has a positive effect on multi-party cooperation.

In the three-way evolutionary game, regardless of the strategy adopted by the government, enterprises and medical institutions will adopt the strategy of producing and using EFMs, but government regulations can promote the green transformation process, as shown in Figure 9.

Figure 10 illustrates the impact of different proportions of the initial value of government regulation. On the one hand, if the government deregulates (or small initial value), the transformation time becomes longer, and businesses will be less enthusiastic about green transformation (blue line and red dashed line in Figure 9 and Figure 10). On the other hand, the continuous use of disposable masks by medical institutions has an adverse impact on the environment (black thin dash-and-dot line and red line in Figure 10). If the government deregulates (or large initial value), the longer the green transformation lasts, and the reputation loss will increase with an increase in pollution. When critical reputation loss is exceeded, the government must intervene and regulate the green transformation process. If the government regulates earlier (large initial value), pollution will be lower (short time), the reputation loss of the government will be lower, and the speed of green transformation will be faster (green dashed line and black dashed line in Figure 10).

Figure 11 illustrates the effect of different price increase coefficients of EFMs on decisions of government (a), enterprises (b), and medical institutions (c). If the price of EFMs is too high, medical institutions may not use them at all (green dashed line in Figure 11c); if the price is too low, the enterprises’ profits will be small or even negative, thereby reducing their enthusiasm for producing EFMs (blue line in Figure 11b). The high price of masks will require the government to keep regulating (green dashed line in Figure 11a).

Figure 12 illustrates the effect of different cost increase coefficients of EFMs on the decisions of government (a), enterprises (b), and medical institutions (c). The cost of EFMs was observed to have a smaller effect on the proportion of medical institutions that used EFMs (Figure 12c). For governments, the lower the cost of EFMs, the greater the probability that the government will choose deregulation (blue line in Figure 12a), and the greater the probability that enterprises will choose to produce EFMs (blue line in Figure 12b).

Figure 13 illustrates the effect of different reputation damages from government inaction on decisions of government (a), enterprises (b), and medical institutions (c). If reputation damage is too high, the government will choose regulation (blue line in Figure 13a). Reputation damage increases and green transformation enthusiasm for medical institutions and enterprises increases (blue line in Figure 13b,c).

Figure 14 illustrates the effect of different cost benefits for using EFMs (Note: This paper compares the large positive, normal positive, zero, large negative, and normal negative cost benefits for using EFMs) on decisions of government (a), enterprises (b), and medical institutions (c). Medical institutions and enterprises will be more willing to produce and use EFMs if the reusability is greater and the cleaning procedure is simpler and cheaper (black thin dash-and-dot line in Figure 14b,c). When the cost of reusing masks is high, the government tends to regulate them (blue line in Figure 14a), while medical institutions may not use EFMs (blue line in Figure 14c). This leads to the government incurring greater regulatory costs to encourage companies and medical institutions to produce and use EFMs.

Figure 15 illustrates the effect of different negative cost benefits for using EFMs with different subsidies on the decisions of medical institutions. This paper found that medical institutions treat green transformation negatively if the cost of using EFMs is the same as, or even higher than, that of disposable masks. In such cases, the green transformation will be hard to complete, regardless of how much the government subsidizes it (blue line in Figure 15). Therefore, this analysis suggested that the government should set aside funds for mask research and development, increase the level of technological innovation in the production of masks, and reduce production costs.

Figure 16 illustrates the effect of different liquidated damages for the termination of medical institutions’ purchase of ordinary masks on the decisions of government (a), enterprises (b), and medical institutions (c). A penalty for terminating the purchase of ordinary masks imposed by enterprises on medical institutions will have a negative impact on the environmental protection choices of the government (green dashed line in Figure 16a), enterprises (green dashed line in Figure 16b), and medical institutions (green dashed line in Figure 16c). Therefore, the government should excel at supervising, coordinating, and adopting appropriate policies to reduce or cancel the liquidated damages for terminating the purchase agreement for ordinary masks between enterprises and medical institutions.

## 5. Results

Through the tripartite evolutionary game, this study examined the behavioral strategies of governments, enterprises, and medical institutions, aiming to promote, produce, and use EFMs. The study obtained the dynamic replicator equations and evolutionary stabilization strategies for different groups, conducted a simulation analysis using system dynamics, and drew the following conclusions:
The model in this study has a unique three-way evolutionary stable equilibrium strategy (government deregulation, enterprises producing EFMs, and medical institutions using EFMs). The government should formulate appropriate rules and regulations to encourage tripartite cooperation for effective epidemic prevention and environmental protection. Furthermore, the government should also actively promote the realization of environmental protection goals without affecting epidemic prevention and control;Through a system dynamics simulation analysis, it is revealed that changes in external parameters have a significant impact on the strategy selection process of all players, as shown in Table 3. H and $f_{1}$ are positively correlated with the proportion of government regulation; $C_{G}$, $G_{C}$, and $G_{H}$ are negatively correlated with the proportion of government regulation. α, P, and $G_{C}$ are positively correlated with the proportion of enterprises that produce EFMs. $f_{2}$, β, and $C_{S}$ are negatively correlated with the proportion of enterprises that produce EFMs. $r_{c}$, $C_{H}$, and $f_{1}$ are positively correlated with the proportion of medical institutions that use EFMs. α, d, and $f_{2}$ are negatively correlated with the proportion of medical institutions that use EFMs;When the cost of using EFMs is the same as, or even higher than, that of disposable masks, medical institutions will treat green transformation negatively and, regardless of government subsidies, it will be hard to complete the green transformation. Therefore, the government should, on the basis of regulations, increase investments in scientific and technological research and development, as well as lower the price and usage cost of EFMs. Moreover, the government should motivate medical institutions to use EFMs, thereby encouraging enterprises to produce EFMs.If there is a penalty for terminating the purchase of ordinary masks for medical institutions and businesses during the transformation period, it can negatively impact the cooperation between the two entities. Consequently, the green transformation process will also be adversely affected.When the reputation damage from government inaction is greater than the cost of regulation, regardless of the initial conditions of evolution, it is always an optimal strategy for the government to regulate the production and use of EFMs. Environmental protection and sustainability are currently among the most important national strategies. People have increasingly higher requirements for the ecological environment, and reputation loss is increasing daily. Therefore, reasonable and feasible regulations should be formulated immediately to realize the green transformation of anti-epidemic supplies.


## 6. Conclusions

The COVID-19 pandemic will continue, and conflicts between medical and environmental protection need to be solved. With the continuous development of EFMs, such as CuMask+, EFL, etc., as the leaders of green transformation, governments, should consider reasonable regulatory measures to promote the production and use of EFMs by enterprises and consumers in order to reduce the contradiction. As a whole, the government should invest management costs and establish a reward and punishment system under the conditions of low manufacturing costs, low price, and low reuse costs in the early and mid-stages of EFMs development. This measure will effectively speed up the green transformation of anti-epidemic supplies. In the mid-transformation period, enterprises and medical institutions will continue to opt for green transformation if the government reduces regulations. In the later stages of the transformation, the government will stop regulating, as enterprises and medical institutions are more likely to produce and use EFMs. Furthermore, there were two interesting findings during the simulation process. On the one hand, liquidated damages imposed by enterprises on medical institutions for terminating the purchase of ordinary masks will have a negative impact on green transformation. Medical institutions, on the other hand, perceive green transformation negatively when the cost of using EFMs is the same as, or even higher than, that of disposable masks. No matter how much the government subsidizes it, green transformation will be hard to complete. Synthesizing the research of this paper, the government should, on the basis of regulations, increase investments in scientific and technological research and development, as well as lower the price and usage cost of EFMs. Moreover, the government should motivate medical institutions to use EFMs, thereby encouraging enterprises to produce EFMs in order to achieve the ultimate goal of green transformation of anti-epidemic supplies in the post-pandemic era.

## 7. Discussion

This paper studied the influence of the three players (government, enterprises, and medical institutions), constructed a replica dynamic system of an asymmetric evolutionary game, and conducted a dynamic simulation analysis. The evolutionary game and system dynamics are used to study the relationship between parameters. By conducting further research on epidemic protection behavior [37,38,39], this paper focused on the economic interests of medical institutions and manufacturers of masks in order to show their actual behavior. On the basis of green transformation studies [43,44,45,46,47,48,49,50], this paper considers consumers (medical institutions) as one of the game subjects and adds parameters related to medical institutions in order to study the impact of medical institutions on green transformation.

Research has shown that the value of external variables directly affects the strategic choices of all players, and the only tripartite evolutionary stability strategy is for governments to deregulate mask production, enterprises to increase eco-friendly mask production, and medical institutions to use them. The optimal strategy for evolution is deregulation. It can be seen that the green transformation of anti-epidemic supplies cannot be achieved solely by relying on government regulations. The best way to achieve it is to promote environmental protection knowledge in various aspects, encourage the use of eco-friendly products, vigorously develop productivity, reduce costs, reduce the production and recycling costs of EFMs, lower the threshold for use and cleaning, and establish a good market environment for EFMs. In addition, it can be seen from the simulation that the production and use of EFMs is a trend in the post-pandemic era. When effective and economical EFMs are developed and produced, epidemic prevention measures undergo green transformation. This further confirms that government regulation has a positive impact on green transformation. This research can help governments, enterprises, and medical institutions to better understand the role of epidemic prevention materials during the green transformation period, as well as serve as a guide for governments in determining the most appropriate regulatory measures.

Considering the need for epidemic prevention, this study did not consider environmental fines levied on enterprises by governments. Therefore, enterprises, as the most direct beneficiaries, have an obligation to undertake the responsibility of environmental protection. Enterprises need to invest in research and development to improve the effectiveness of EFMs while considering their own interests. This will encourage medical institutions to consistently purchase and use EFMs and facilitate green transformation. Medical institutions, as the primary practitioners of the green transformation of anti-epidemic supplies, must approach EFMs and other products with optimism, and actively explore ways to make the healthcare sector environmentally sustainable.

The simulation research in this study is closely related to the selection of the initial values of the exogenous variables. When the relevant policies are changed and adjusted, the results of each parameter will show different trends. Therefore, future research must establish different models for flexible policies. Moreover, this study does not consider the impact of the supply and demand of new EFMs on the price of existing ones in the market, as well as the impact of consumer preferences on the EFMs market. These factors must be considered in future studies.

## 8. Code Availability

The system dynamics model was implemented using Vensim PLE for Windows Version 7.3.5 (Harvard, MA, USA) to draw the flow chart of the system and simulation analysis. Specific operational instructions are available at https://vensim.com/support/ (accessed on 9 April 2022).

## Figures and Tables

**Figure 1 ijerph-19-06011-f001:**
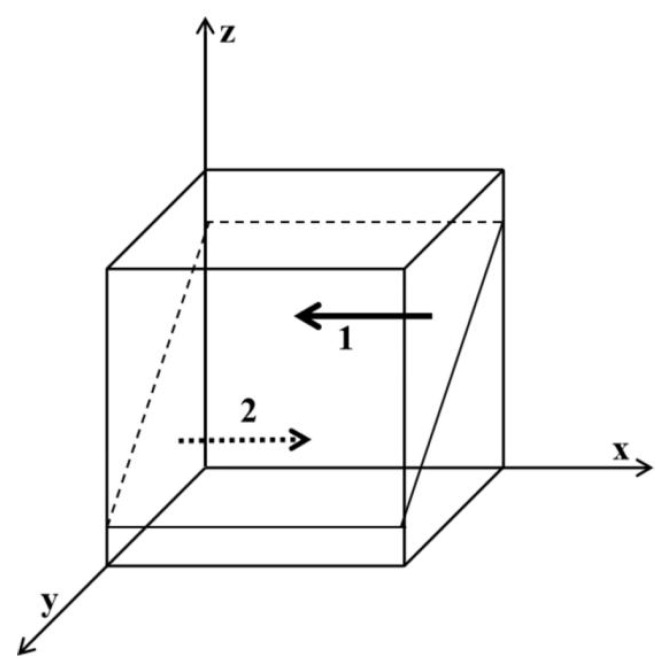
Influence of z on the evolutionary equilibrium stable strategy of x. Region 1 is no regulation; region 2 is regulation.

**Figure 2 ijerph-19-06011-f002:**
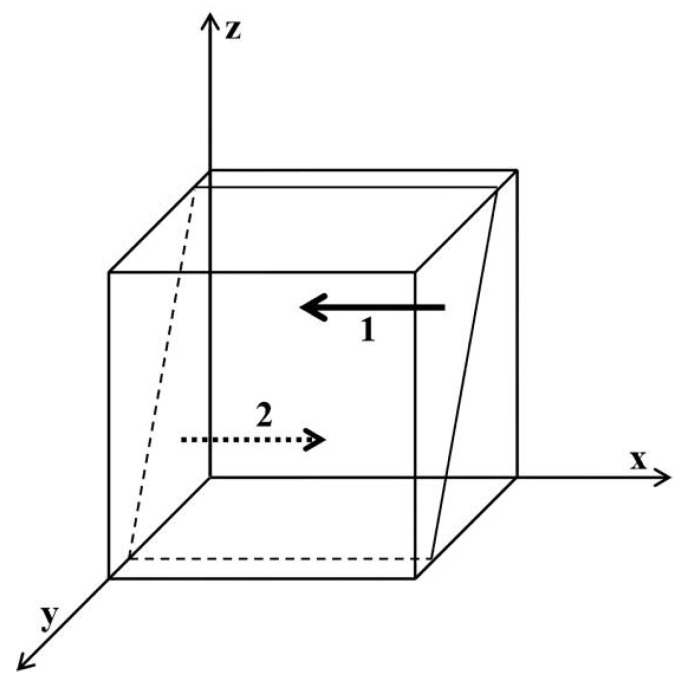
Influence of y on the evolutionary equilibrium stable strategy of x. Region 1 is no regulation; region 2 is regulation.

**Figure 3 ijerph-19-06011-f003:**
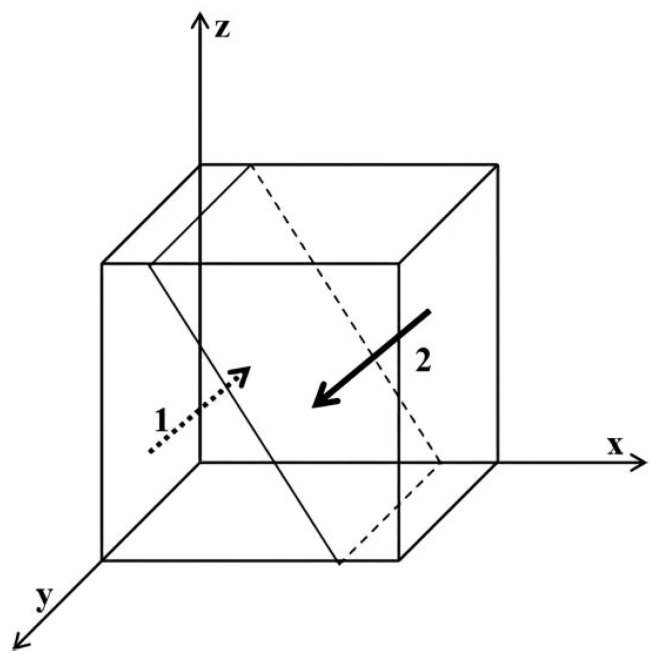
Influence of x on the evolutionary equilibrium stable strategy of y. Region 1 is no increase in production; region 2 is increase in production.

**Figure 4 ijerph-19-06011-f004:**
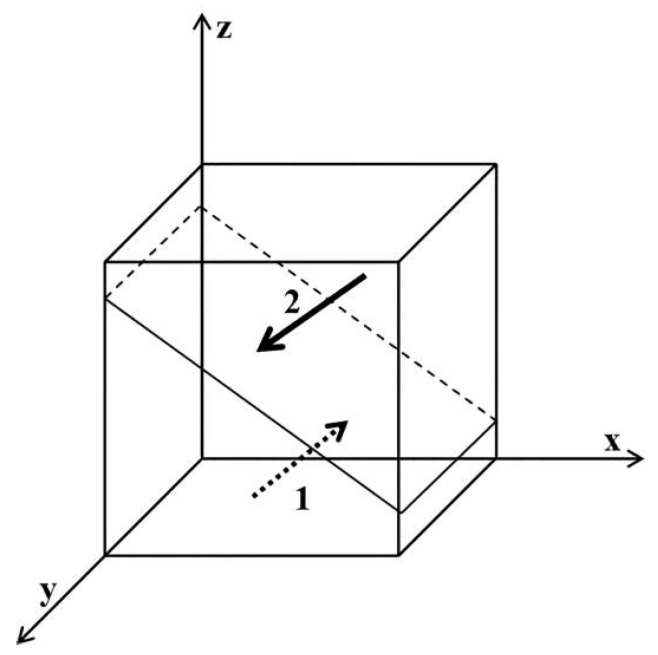
Influence of z on the evolutionary equilibrium stable strategy of y. Region 1 is no increase in production; region 2 is increase in production.

**Figure 5 ijerph-19-06011-f005:**
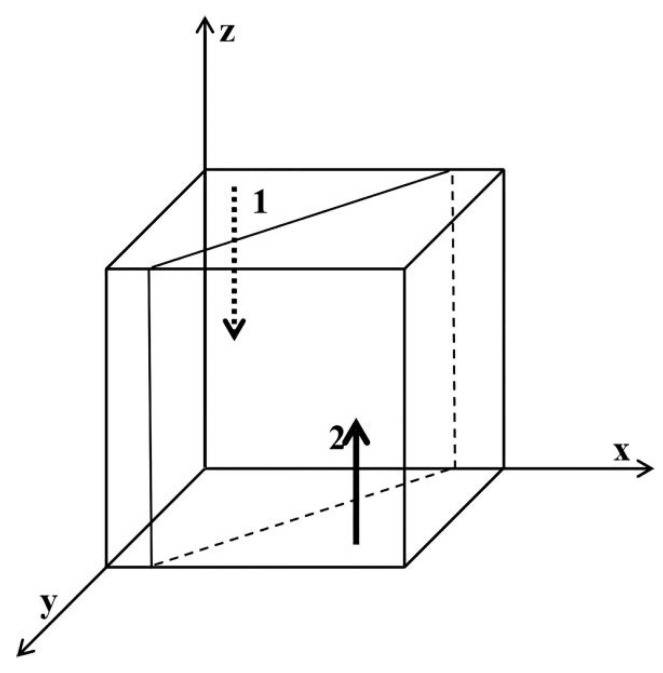
Influence of x on the evolutionary equilibrium stable strategy of z. Region 1 is using regular masks; region 2 is using EFMs.

**Figure 6 ijerph-19-06011-f006:**
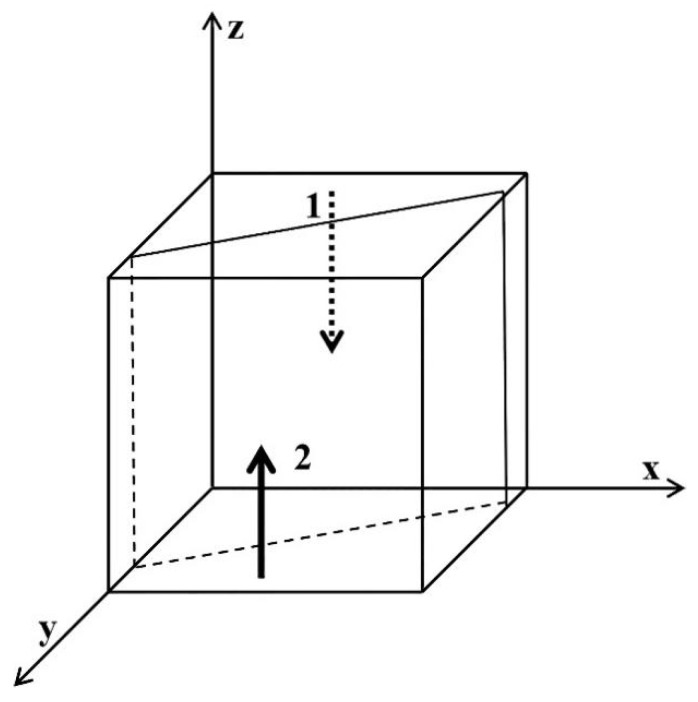
Influence of y on the evolutionary equilibrium stable strategy of z. Region 1 is using regular masks; region 2 is using EFMs.

**Figure 7 ijerph-19-06011-f007:**
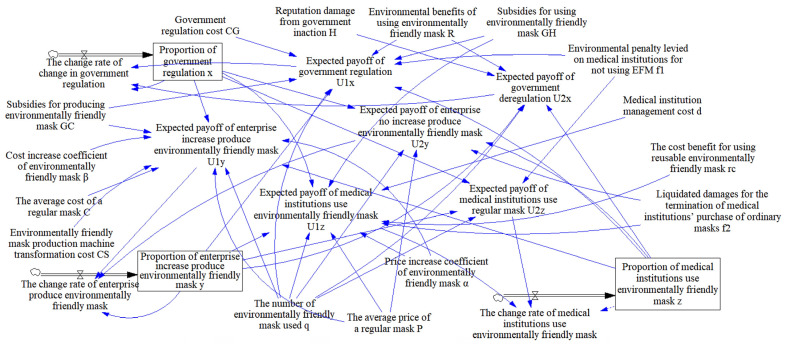
The system dynamics model of governments, enterprises, and medical institutions.

**Figure 8 ijerph-19-06011-f008:**
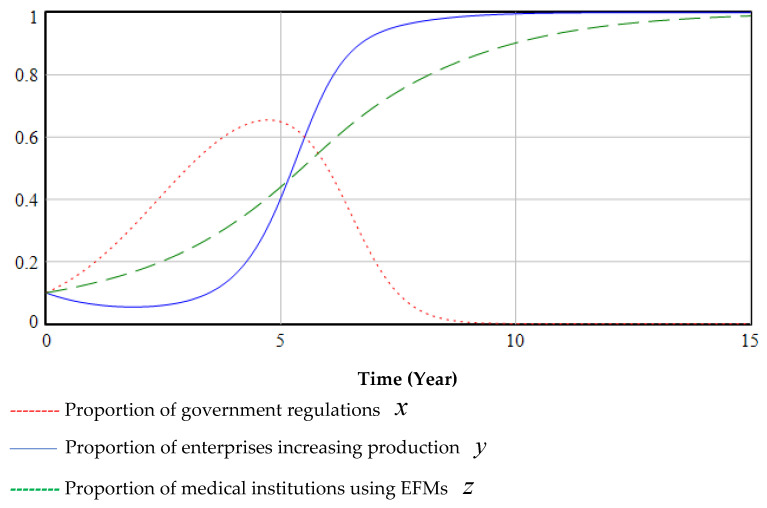
Stability test of ESS point (0, 1, 1).

**Figure 9 ijerph-19-06011-f009:**
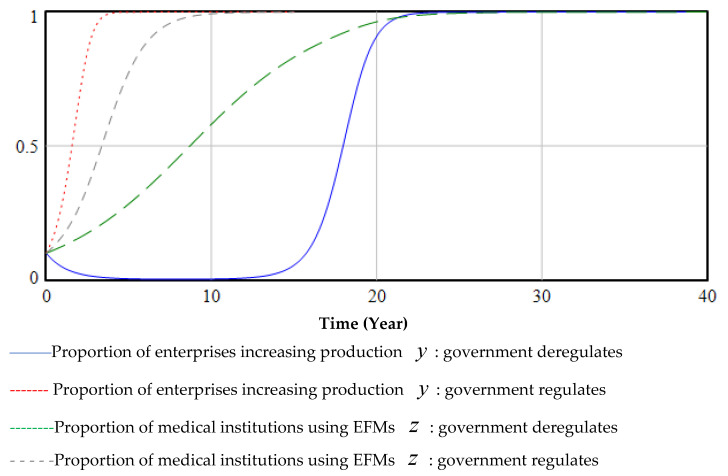
The impact of government regulation on the decisions of hospitals and enterprises.

**Figure 10 ijerph-19-06011-f010:**
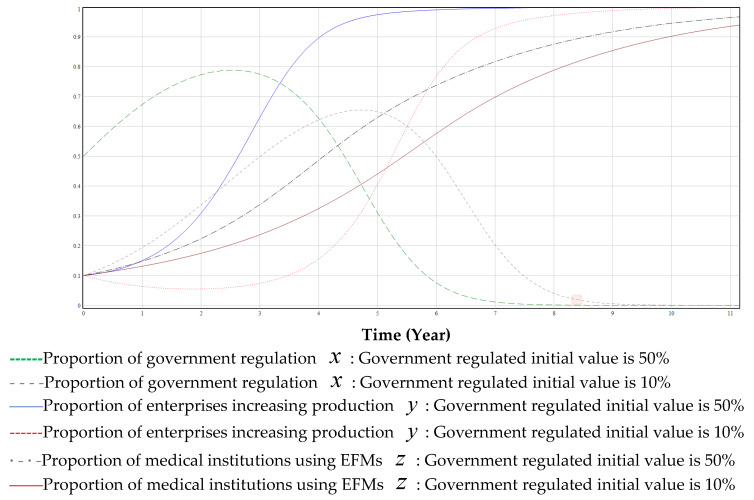
The impact of a high proportion of government regulation (initial value of 50% compared to 10%).

**Figure 11 ijerph-19-06011-f011:**
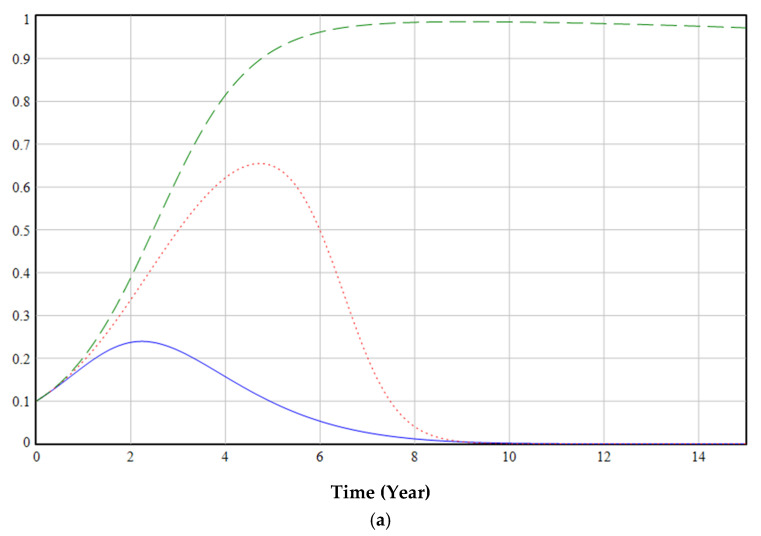
The effect of α on decisions of government (**a**), enterprises (**b**), and medical institutions (**c**).

**Figure 12 ijerph-19-06011-f012:**
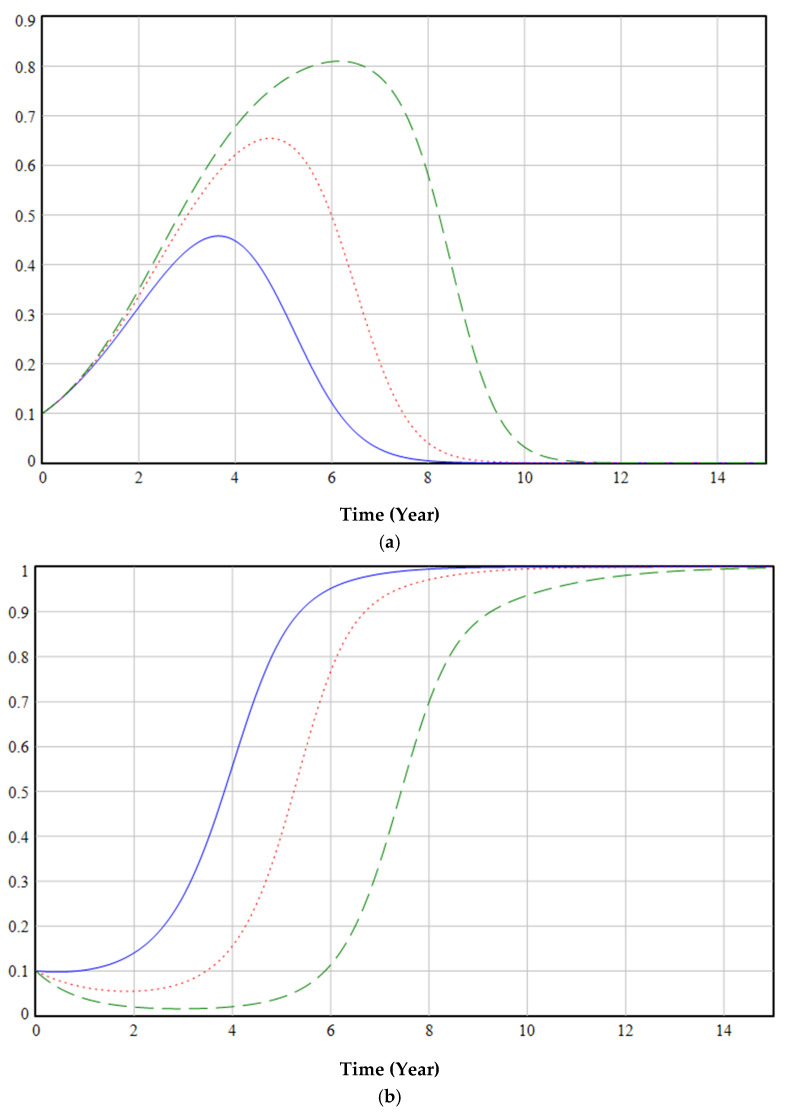
The effect of β on decisions of government (**a**), enterprises (**b**), and medical institutions (**c**).

**Figure 13 ijerph-19-06011-f013:**
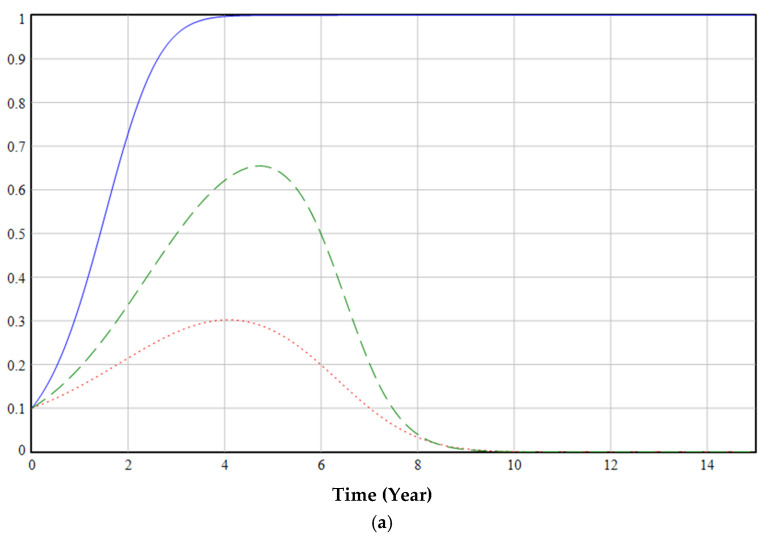
The effect of H on decisions of government (**a**), enterprises (**b**), and medical institutions (**c**).

**Figure 14 ijerph-19-06011-f014:**
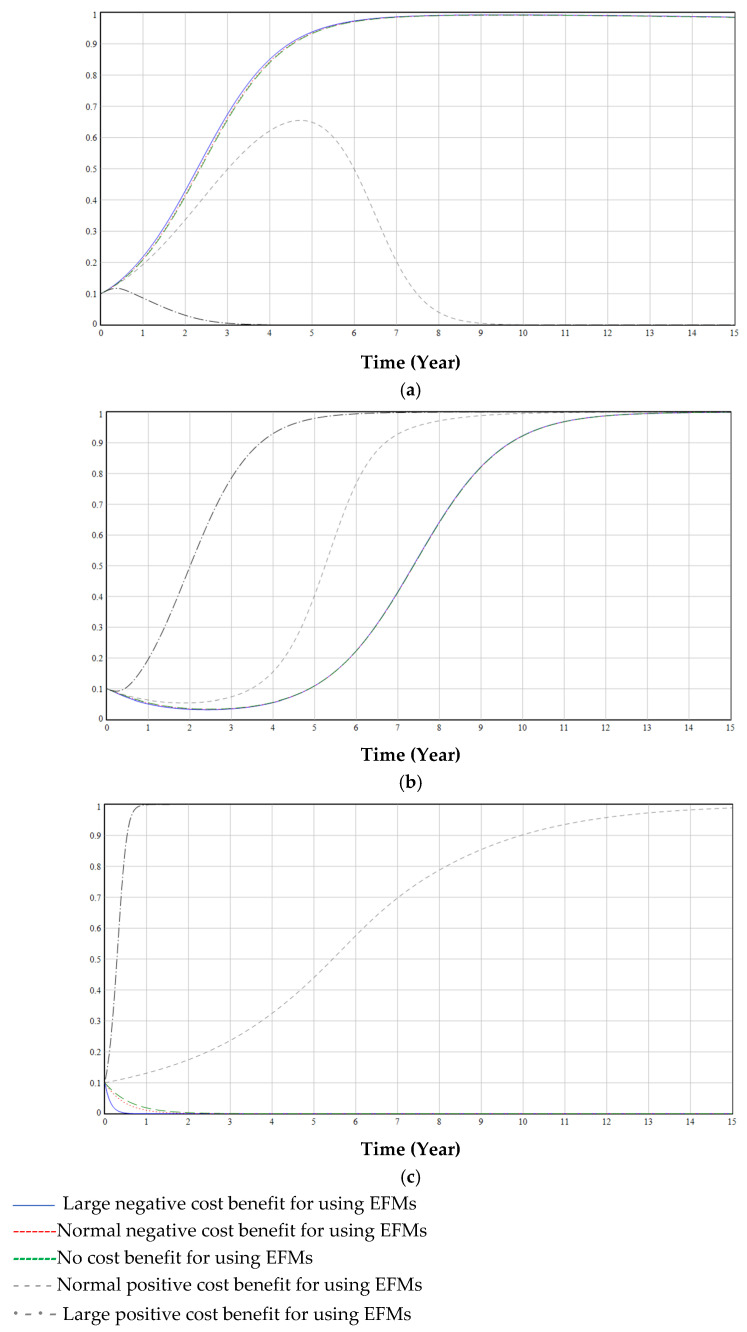
The effect of $r_{c}$ on decisions of government (**a**), enterprises (**b**), and medical institutions (**c**).

**Figure 15 ijerph-19-06011-f015:**
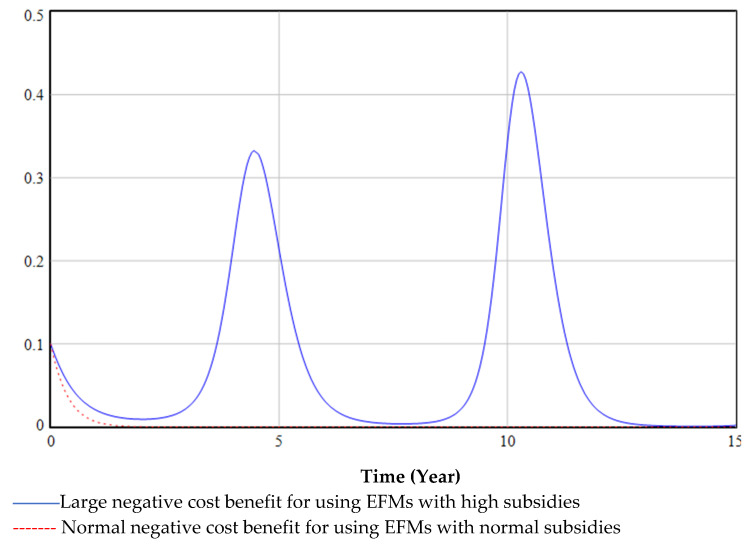
Different negative cost benefits for using EFMs with different subsidies.

**Figure 16 ijerph-19-06011-f016:**
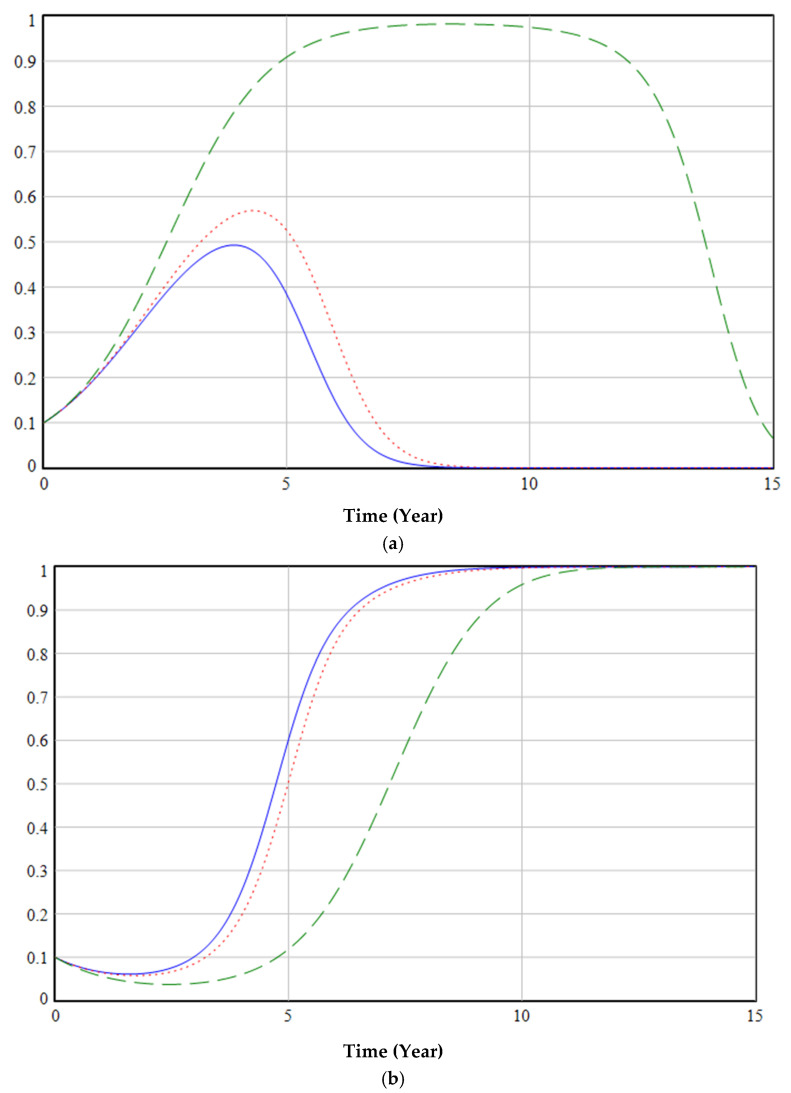
The effect of $f_{2}$ on decisions of government (**a**), enterprises (**b**), and medical institutions (**c**).

**Table 1 ijerph-19-06011-t001:** Model variables.

Variable Name	Definition	Units	Initial Value
R	Environmental benefits of using EFMs	CNY/PC	0.8
H	Reputation damage from government inaction	CNY/PC	1.8
$G_{C}$	Subsidies for producing EFMs	CNY/PC	2
$G_{H}$	Subsidies for using EFMs	CNY/PC	0.2
P	The average price of a regular mask	CNY/PC	1
C	The average cost of a regular mask	CNY/PC	0.5
$C_{G}$	Government regulation cost	CNY/PC	0.05
$C_{S}$	EFM production machine transformation cost	CNY/PC	0.3
d	Medical institution management cost	CNY/PC	0.05
$r_{c}$	The cost benefit for using EFMs	CNY/PC	2
$f_{1}$	Environmental penalty levied on medical institutions for not using EFMs	CNY/PC	0.1
$f_{2}$	Liquidated damages for the termination of medical institutions’ purchase of ordinary masks	CNY/PC	0.2
q	The number of EFMs used	PC	1
α	Price increase coefficient of EFMs		1.5
β	Cost increase coefficient of EFMs		1.5

Note: The initial values are set up, with some modifications, according to the Chinese mask market. The initial values of the variables are as close to reality as possible. For exogenous variables lacking a realistic reference, such as punishment, reputation loss, and social and environmental benefits, the principle of positive benefits of the three players should be guaranteed.

**Table 2 ijerph-19-06011-t002:** Evolutionary game payoff matrix.

	**Government Regulation** (x)	
	**Medical Institutions Using EFMs** (z)	**Medical Institutions Using Regular Masks** (1−z)
**Enterprises increasing the production of EFMs** (y)	$\left(R-G_{c}-G_{H}-C_{G}\right)q$	$\left(f_{1}-G_{c}-G_{G}\right)q$
	$\left(\alpha P+G_{c}-\beta C-C_{S}\right)q$	$\left(G_{c}-\beta C-C_{S}\right)q$
	$\left(G_{H}-\alpha P-d+r_{c}\right)q$	$-f_{1}q$
**Enterprises not increasing the production of EFMs** (1−y)	$\left(R-G_{H}-C_{G}\right)q$	$\left(f_{1}-C_{G}\right)q$
	$\left(f_{2}-P\right)q$	0
	$\left(G_{H}-\alpha P-d+r_{c}+f_{2}\right)q$	$-f_{1}q$
	**Government Deregulation** (1−x)	
	**Medical Institutions Using EFMs** (z)	**Medical Institutions Using Regular Masks** (1−z)
**Enterprises increasing the production of EFMs** (y)	(R−H)q	−Hq
	$\left(\alpha P-\beta C-C_{S}\right)q$	$\left(-\beta C-C_{S}\right)q$
	$\left(-\alpha P-d+r_{c}\right)q$	0
**Enterprises not increasing the production of EFMs** (1−y)	(R−H)q	−Hq
	$\left(f_{2}-P\right)q$	0
	$\left(-\alpha P-d+r_{c}-f_{2}\right)q$	0

**Table 3 ijerph-19-06011-t003:** Parameter influence.

Decision	Prove	Variable Name	Correlation
Government regulation	$\frac{dV_{11}}{dH}=\frac{1}{f_{1}+G_{H}}>0$	H	+
	$\frac{dV_{11}}{dC_{G}}=\frac{-1}{f_{1}+G_{H}}<0$	$C_{G}$	−
	$\frac{dV_{11}}{dG_{C}}=\frac{-1}{2f_{1}+2G_{H}}<0$	$G_{C}$	−
	$\frac{dV_{12}}{dG_{H}}=\frac{-1}{2G_{C}}<0$	$G_{H}$	−
	$\frac{dV_{12}}{df_{1}}=\frac{1}{2G_{C}}<0$	$f_{1}$	+
Enterprises increasing the production of EFMs	$\frac{dW_{21}}{df_{2}}=\frac{-1}{2G_{C}}<0$	$f_{2}$	−
	$\frac{dW_{21}}{d\beta C}=\frac{-1}{G_{C}}<0$	β	−
	$\frac{dW_{21}}{dC_{S}}=\frac{-1}{2G_{C}}<0$	$C_{S}$	−
	$\frac{dW_{21}}{d\alpha P}=\frac{1}{2G_{C}}>0$	α	+
	$\frac{dW_{21}}{dP}=\frac{1}{2G_{C}}>0$	P	+
	$\frac{dW_{22}}{dG_{C}}=\frac{1}{2\alpha P-2f_{2}}>0$	$G_{C}$	+
Medical institutions using EFMs	$\frac{dW_{31}}{d\alpha P}=\frac{-1}{G_{H}+f_{1}}>0$	α	−
	$\frac{dW_{31}}{dd}=\frac{-1}{G_{H}+f_{1}}<0$	d	−
	$\frac{dW_{31}}{dr_{c}}=\frac{1}{G_{H}+f_{1}}<0$	$r_{c}$	+
	$\frac{dW_{31}}{df_{2}}=\frac{-1}{2G_{H}+2f_{1}}<0$	$f_{2}$	−
	$\frac{dW_{32}}{dG_{H}}=\frac{1}{2f_{2}}>0$	$G_{H}$	+
	$\frac{dW_{32}}{df_{1}}=\frac{1}{2f_{2}}>0$	$f_{1}$	+

Note: + is positive correlation; − is negative correlation.

**Table 4 ijerph-19-06011-t004:** Attribute analysis of evolutionary equilibrium points.

Equilibrium Point	Eigenvalues			Attributes
$O_{1}$(0, 0, 0)	1.85	−0.05	0.25	Saddle point
$O_{2}$(1, 0, 0)	−1.85	1.95	0.55	Saddle point
$O_{3}$(0, 1, 0)	−0.15	0.05	0.45	Saddle point
$O_{4}$(0, 0, 1)	1.55	1.25	−0.25	Saddle point
$O_{5}$(1, 1, 0)	0.15	−1.95	0.75	Saddle point
$O_{6}$(1, 0, 1)	−1.55	3.25	−0.55	Saddle point
$O_{7}$(0, 1, 1)	−0.45	−1.25	−0.45	Stable point
$O_{8}$(1, 1, 1)	0.45	−3.25	−0.75	Saddle point

## Data Availability

Not applicable.

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
