# Peer review of "Green Transformation of Anti-Epidemic Supplies in the Post-Pandemic Era: An Evolutionary Approach"

_ijerph, 2022, doi:10.3390/ijerph19106011_

Round 1

Reviewer 1 Report

ijerph-1698229-peer-review-v1

Review of “ Green Transformation of Anti-Epidemic Supplies…”

In their problem setting, the authors may also want to consider dispersed effects in mask pollution as well as masks not discarded as medical waste.

The paper is convincingly written and appears methodologically sound.

Section 5 presents the results rather than Conclusions

A proper discussion section, which ties the findings back to the literature in the introduction section and anchors the meaning of the findings in the preexisting research is lacking.

The wording ‘recommendations’ should be removed and the implications more fleshed out. This is not well enough developed and as it stands the authors do injustice to their own work

Minor issues

The systems dynamics model needs to be displayed larger as it is too hard to read

Author Response

Response to Reviewer 1 Comments

We would first like to thank you for all the constructive and critical comments on our paper. When revising this paper, we would like to assure you that we have seriously considered all comments and diligently addressed all concerns as much as possible. The changes are highlighted in yellow in the new manuscript. Regarding your comments (in italics), we have addressed them in our revision as follows. Thank you very much!

Point 1: Section 5 presents the results rather than Conclusions 

Response 1: Thanks for your valuable comments. Indeed, Section 5 presents the Results rather than Conclusions. We changed the title of the Conclusions into Results, and then combined with Point 2 to add the content of the Conclusions

Point 2: A proper discussion section, which ties the findings back to the literature in the introduction section and anchors the meaning of the findings in the preexisting research is lacking.

Response 2: Thanks for your constructive comments. As mentioned in Point 1, we added a new Conclusion section and modified the Discussion section. According to your comments, we have further analyzed and explained the Conclusion and Discussion section, and also put forward more effective managerial insights and policy suggestions. The new Discussion section better combines conclusion and introduction. For example, we have supplemented this paragraph “Conducted further research on epidemic protection behavior ...... considers consumers (medical institutions) as one of the game subjects and adds parameters related to medical institutions in order to study the impact of medical institutions on green transformation.”

Point 3: The wording ‘recommendations’ should be removed and the implications more fleshed out. This is not well enough developed and as it stands the authors do injustice to their own work

Response 3: Thanks for your objective and critical comments. We are aware of the issue and have removed The wording ‘recommendations’.

Point 4: The systems dynamics model needs to be displayed larger as it is too hard to read

Response 4: Thanks for raising this issue. We adjusted the font size and the figure size to increase readability in the reader's perception. In addition, the resolution of Figure. 7 has been optimized.

Before closing, we wish to thank you again in helping us to improve the quality of this paper. Thank you!

Reviewer 2 Report

What is the main question in the study?

The main question presented by the authors is:

How to effectively manage the green crossing to alleviate the use of medical supplies, such as masks, to adequately reduce environmental problems during the COVID-19 pandemic. In the article, the authors answer three questions:

- what impact do government regulations have on the green transformation,

- how companies and medical institutions make decisions in the face of government regulations,

- what are the factors influencing the green transformation.

Is it relevant and interesting?

It is currently a particularly interesting issue, because the world is struggling with a significant problem during the pandemic, which is the appropriate management of medical resources used in the fight against the pandemic, and in particular limiting the negative impact of these funds on the quality of the natural environment.

How original is the text?

The text is interesting and original, as it touches upon issues of great importance to the whole world related to the protection of the environment and, consequently, the protection of the health of the society. This article presents the green transformation of antiepidemic supplies in the post-pandemic era, using the method of analysis and establishing an evolutionary game model, involving governments, companies producing anti-epidemic products and medical institutions.

What does this add to the comparable subject area?

The article is an important contribution to showing the impact of environmental regulations on green transition in order to mitigate environmental degradation caused by the rapid increase in the amount of medical waste. Particular attention was paid to the use of reusable masks.

With other published material?

In the discussion, practically no reference was made to the results of similar research conducted around the world. These results are to some extent presented in the introduction.

Is the paper well written?

The article is written correctly.

Is the text clear and easy to read?

The text is clear and accessible to the reader.

Are the conclusions consistent with the evidence and arguments presented?

Conclusions and recommendations are in line with the results of the conducted analyzes.

Do they relate to the main question posed?

The conclusions refer to the questions posed in the article.

Figure 7 is unreadable, you may consider enlarging it (in terms of horizontal page orientation) to increase readability in the reader's perception.

Author Response

Response to Reviewer 2 Comments

We would first like to thank you for all the constructive and critical comments on our paper. When revising this paper, we would like to assure you that we have seriously considered all comments and diligently addressed all concerns as much as possible. The changes are highlighted in yellow in the new manuscript. Regarding your comments (in italics), we have addressed them in our revision as follows. Thank you very much!

Point 1: Figure 7 is unreadable, you may consider enlarging it (in terms of horizontal page orientation) to increase readability in the reader's perception.

Response 1: Thanks for raising this issue. We adjusted the font size and the figure size to increase readability in the reader's perception. In addition, the resolution of Figure. 7 has been optimized.

Before closing, we wish to thank you again in helping us to improve the quality of this paper. Thank you!

Reviewer 3 Report

In "Green Transformation of Anti-Epidemic Supplies in the Post-Pandemic Era: An Evolutionary Approach" authors study a critically relevant and timely problem, and they provide innovative insights that help us better understand and predict the fallout of the COVID-19 pandemic in terms of the footprint of anti-epidemic supplies on the green transformation.

I recommend a revision before further consideration, along the following comment that should be taken into account with care and love to detail.

1) The abstract is a little thin and does not quite convey the vibrancy of the findings and the depth of the main conclusions. The authors should please extend this somewhat for a better first impression.

2) I would also kindly ask to cite the very relevant research paper on solving COVID-19 problems: The benefits, costs and feasibility of a low incidence COVID-19 strategy, Thomas Czypionka et al., Lancet Reg. Health Eur. 13, 100294 (2022) and Risk assessment of COVID-19 epidemic resurgence in relation to SARS-CoV-2 variants and vaccination passes, Tyll Krüger et al., Commun. Med. 2, 23 (2022). These are key references that help shed further light on the post-pandemic state of COVID-19, also in terms of resources we may need in future iterations and their availability and impact on the environment.

3) It would also improve the paper if the figure captions would be made more self contained. In addition to briefly stating what is shown, one could also consider a sentence or two saying what is the main message of each figure.

4) It would be very useful if the authors would make their source code available as supplementary material. This would promote the usage of the proposed model and allow also others to take advantage of this research, and also to allow them to reproduce the results.

5) The presentation of the results in terms of figures is quite poor. The figures have small labels and are hardly adequately presented in terms of labels and what different lines represent. This needs to be improved. The presentation in general is also not in keeping with an interdisciplinary readership. Too much technical details is presented without much guidance of the reader through what is shown and why. This needs improvement for better clarify of the presentation.

6) Some references contain errors, missing or incorrect information, and inconsistent formatting. It is difficult to give credit to research if such elementary aspects of the work are not error free. References should thus be corrected with the best care.

If a revision will be granted, I would be happy to review the manuscript again.

Author Response

Response to Reviewer 3 Comments

We would first like to thank you for all the constructive and critical comments on our paper. When revising this paper, we would like to assure you that we have seriously considered all comments and diligently addressed all concerns as much as possible. The changes are highlighted in yellow in the new manuscript. Regarding your comments (in italics), we have addressed them in our revision as follows. Thank you very much!

Point 1:  The abstract is a little thin and does not quite convey the vibrancy of the findings and the depth of the main conclusions. The authors should please extend this somewhat for a better first impression.

Response 1: Thanks for your constructive comments. According to your comments, we have further extend the abstract and the main conclusions are deepened.  Some main conclusions are explained in more detail. Such as “From the comprehensive analysis, a few important findings are obtained. Firstly, ...... Government should realize the green transformation of anti-epidemic supplies at the earliest in order to avoid severe reputation damage.”

Point 2: I would also kindly ask to cite the very relevant research paper on solving COVID-19 problems: The benefits, costs and feasibility of a low incidence COVID-19 strategy, Thomas Czypionka et al., Lancet Reg. Health Eur. 13, 100294 (2022) and Risk assessment of COVID-19 epidemic resurgence in relation to SARS-CoV-2 variants and vaccination passes, Tyll Krüger et al., Commun. Med. 2, 23 (2022). These are key references that help shed further light on the post-pandemic state of COVID-19, also in terms of resources we may need in future iterations and their availability and impact on the environment.

Response 2: Thanks for your valuable comments. The articles you have provided have been of great help to us. On the one hand these key references provide us COVID-19 is continuous and iterative. On the other hand, was also provide us the feasibility demonstration of non-pharmaceutical interventions is feasible.

Point 3: It would also improve the paper if the figure captions would be made more self contained. In addition to briefly stating what is shown, one could also consider a sentence or two saying what is the main message of each figure.

Response 3: Thanks for your valuable and constructive comments. According to your comments, we have further analyzed and explained these figures. Combined with Point 5, we further explained the figure and put the explanation as close as possible to the picture. The curves corresponding to the results were also marked. Such as “If the curve is closer to 1, it means that the probability of selecting regulation or increasing the production of EFMs or using EFMs is higher. Vice versa.”, “Even if the government cancels relevant regulatory policies at a later stage, medical institutions and enterprises will evolve to use and produce EFMs (Blue line and green dashed line in Figure 8).” and “Figure 13 illustrates the effect of different reputation damage from government inaction on decisions of government (a), enterprises (b) and medical institutions (c).”

Point 4: It would be very useful if the authors would make their source code available as supplementary material. This would promote the usage of the proposed model and allow also others to take advantage of this research, and also to allow them to reproduce the results.

Response 4: Thanks for your kind comments. Software and code have been added to the article. The code uses packages within the software, so we added the manual to the software

Point 5: The presentation of the results in terms of figures is quite poor. The figures have small labels and are hardly adequately presented in terms of labels and what different lines represent. This needs to be improved. The presentation in general is also not in keeping with an interdisciplinary readership. Too much technical details is presented without much guidance of the reader through what is shown and why. This needs improvement for better clarify of the presentation.

Response 5: Thanks for your critical and constructive comments. We have changed the presentation of figures. As mentioned in Point 3, In order to make it easier for interdisciplinary researchers to understand, more obvious guidance and instructions have been added to the paper.

Point 6: Some references contain errors, missing or incorrect information, and inconsistent formatting. It is difficult to give credit to research if such elementary aspects of the work are not error free. References should thus be corrected with the best care.

Response 6: Thanks for raising this issue. We have identified that some of the references is not standard enough and made changes accordingly.

Before closing, we wish to thank you again in helping us to improve the quality of this paper. Thank you!

Reviewer 4 Report

The structure of the paper can be rearranged based on the journal guidelines!

The rest is fine; it reads well, is constructed in reasonably assessed hypotheses, and uses a reasonable methodology to evaluate the hyphothesis.

Author Response

Response to Reviewer 4 Comments

We would first like to thank you for all the constructive and critical comments on our paper. When revising this paper, we would like to assure you that we have seriously considered all comments and diligently addressed all concerns as much as possible. The changes are highlighted in yellow in the new manuscript. Regarding your comments (in italics), we have addressed them in our revision as follows. Thank you very much!

Point 1: The structure of the paper can be rearranged based on the journal guidelines!

Response 1: Thanks for your valuable comments. We have completely revised structure of the paper based on the journal guidelines. In addition, we enlarged the pictures and added the corresponding explanation in the article.

Before closing, we wish to thank you again in helping us to improve the quality of this paper. Thank you!

Round 2

Reviewer 1 Report

The authors have adequately addressed my concerns

Reviewer 3 Report

The authors have revised their manuscript comprehensively and with love to detail. I warmly recommend publication in present form.